# Temp-SCONE: A Novel Out-of-Distribution Detection and Domain Generalization Framework for Wild Data with Temporal Shift

**Aditi Naiknaware*** **Sanchit Singh*** **Hajar Homayouni** **Salimeh Sekeh**

Department of Computer Science, San Diego State University
{anaiknaware7153, ssingh1949, hhomayouni, ssekeh}@sdsu.edu,     *: Equal contribution

## Abstract

Open-world learning (OWL) requires models that can adapt to evolving environments while reliably detecting out-of-distribution (OOD) inputs. Existing approaches, such as SCONE, achieve robustness to covariate and semantic shifts but assume static environments, leading to degraded performance in dynamic domains. In this paper, we propose *Temp-SCONE*, a temporally-consistent extension of SCONE designed to handle temporal shifts in dynamic environments. Temp-SCONE introduces a confidence-driven regularization loss based on Average Thresholded Confidence (ATC), penalizing instability in predictions across time steps while preserving SCONE's energy-margin separation. Experiments on dynamic datasets demonstrate that Temp-SCONE significantly improves robustness under temporal drift, yielding higher corrupted-data accuracy and more reliable OOD detection compared to SCONE. On distinct datasets without temporal continuity, Temp-SCONE maintains comparable performance, highlighting the importance and limitations of temporal regularization. Our theoretical insights on temporal stability and generalization error further establish Temp-SCONE as a step toward reliable OWL in evolving dynamic environments.

## 1 Introduction

Reliable open-world learning (OWL) for Artificial Intelligence (AI) provides a paradigm where AI models learn and adapt to a dynamic-world assumption such that agents encounter unexpected environments Zhu et al. [2024]. Machine learning (ML) models deployed in real-world environments inevitably encounter data that differs from the training distribution. For example, a simple cat-vs-dog classifier trained on curated datasets may, once deployed, receive an input image of an elephant. Since such an input lies outside the model's training distribution, the model's predictions become unreliable. This challenge is broadly studied under the framework of Out-of-Distribution (OOD) detection Liu et al. [2021], Wang et al. [2022], Park et al. [2021], Tamang et al. [2025], Yang et al. [2024]. Unlike ML models, where the models are trained on seen (in-domain) environments, modern AI agents require detecting and adapting to unseen data and abrupt domain shifts. OWL aims to build a robust human-like system that can transfer and consolidate knowledge incrementally during deployment while adapting to shifted domains and detecting OOD samples. An OWL paradigm on wild data Katz-Samuels et al. [2022] is built upon two parts, unknown rejection (OOD detection), novel class discovery (distribution shift generalization) under dynamic domains. Within OWL context, In-distribution (ID) refers to data drawn from the same distribution as the training set—the data that the model is expected to handle reliably. Prior work in both OOD detection and distribution shift has primarily focused on two categories: (1) covariate shift refers to inputs that belong to the same label

39th Conference on Neural Information Processing Systems (NeurIPS 2025) Workshop: Reliable ML from Unreliable Data.

space as the training data but differ due to changes in the input distribution Ye et al. [2022], Koh et al. [2021]. For example, in autonomous driving, a model trained on ID data with sunny weather may experience a covariate shift when deployed in snowy weather. Similarly, in image classification, a dog image turned upside down or corrupted with Gaussian noise remains labeled as "dog", yet such covariate perturbations can degrade model performance. (2) Semantic shifts occur when entirely new classes are introduced at test time Yang et al. [2024], Ye et al. [2022], such as a classifier trained on cats and dogs encountering an elephant. While these perspectives have significantly advanced both OOD detection and OOD generalization Bai et al. [2023], but they largely overlook temporal dynamics, the fact that data distributions may evolve over time due to changing environments, user behavior, or data sources Yao et al. [2022]. Such temporal shifts can lead to gradual but systematic degradation of model performance if left unchecked. For example, a perception system trained on traffic patterns from one year may underperform as new road constructions, seasonal changes, or evolving driving behaviors shift the data distribution over time.

In this paper, we situate these challenges within the broader paradigm of OWL, where AI systems must not only detect semantic novelty but also adapt to distribution shifts encountered over time "in the wild". We introduce a unified approach that simultaneously generalizes to covariate and temporal shifts while robustly detecting semantic shifts. To characterize temporal drift, we leverage metrics such as average threshold confidence (ATC) Garg and Balakrishnan [2022]( and average confidence (AC)), showing that persistent deviations in these metrics provide strong signals of temporal instability. We evaluate our approach on both static benchmark datasets and dynamic datasets that evolve over time, demonstrating improved robustness under open-world conditions. Among established OOD detection and semantic shift generalization methods, the most recent framework SCONE Bai et al. [2023] learns a robust classifier that detects semantic OOD inputs and generalizes to covariate-OOD data.

**SCONE explanation Bai et al. [2023]:** Consider wild data where the static agent encounters covariate and semantic shifts with distribution $\mathbf{P}_{\text{wild}}$ in (1), where $type = \text{semantic, covariate}$. SCONE is a unified energy margin–based learning framework that leverages freely available unlabeled data in the wild, capturing test-time OOD distributions under both covariate and semantic shifts. By marginalizing the energy function, SCONE enforces a sufficient margin between the OOD detector and ID data, thereby improving the performance of both the classifier $f_\theta$ and detector $g_\theta$.

**SCONE Limitations:** A central limitation of SCONE is its reliance on static environments, while OWL inherently involves dynamic domains. Although the authors report strong performance, our experiments demonstrate that SCONE suffers significant performance degradation when transitioning to new domains. This motivates the following critical yet underexplored hypothesis:

> **Hypothesis:** *Exploiting temporal-based confidence in SCONE improves the OOD generalization in downstream time steps and controls the shocks during domain transition in dynamic environments leading one step towards reliable OWL.*

Toward the hypothesis above, we propose *Temp-SCONE*, a temporally-consistent extension of SCONE designed for dynamic domains. Temp-SCONE builds on SCONE's energy margin–based framework by introducing a temporal regularization loss that stabilizes model confidence across evolving distributions. The method leverages ATC (and AC) to monitor prediction stability on both ID and covariate-shifted samples. When confidence drift between consecutive timesteps exceeds a tolerance, Temp-SCONE applies a differentiable temporal loss with adaptive weighting, penalizing instability while preserving flexibility in gradual shifts. This temporal regularization is jointly optimized with cross-entropy and energy-based OOD objectives, allowing Temp-SCONE to maintain strong ID performance while improving robustness to covariate shifts and enhancing semantic OOD detection under dynamic open-world conditions.

**Our main contributions:** We propose Temp-SCONE, a framework for dynamic OOD detection and generalization under temporal shifts. We design a temporal regularization loss using ATC (and AC) to stabilize confidence across time. We demonstrate Temp-SCONE's effectiveness on dynamic (CLEAR, YearBook) and distinct (CIFAR-10, Imagenette, CINIC-10, STL-10) datasets. We provide theoretical insights linking temporal consistency to generalization error bound.

## 2 Methodology

We start with preliminaries to lay the necessary context, followed by our proposed Temp-SCONE method (Section 2.1) and a clear description of SCONE and Temp-SCONE differences.

**Preliminaries:** We consider a deployed classifier $f_\theta : \mathcal{X} \to \mathbb{R}^K$ trained on a labeled in-distribution (ID) dataset $\mathcal{D}_{\text{ID}} = \{(x_i, y_i)\}_{i=1}^n$, drawn *i.i.d.* from the joint data distribution $\mathbb{P}_{\mathcal{X}\mathcal{Y}}$. The function $f_\theta$ predicts the label of an input sample $\mathbf{x}$ as $\hat{y}(f(\mathbf{x})) := \arg\max_y f^y(\mathbf{x})$. Define $\mathbb{P}_{\text{in}}$, the marginal distribution of the labeled data $(\mathcal{X}, \mathcal{Y})$, which is also referred to as the in-distribution. $\mathbb{P}_{\text{out}}^{type}$ is the marginal distribution out of $\mathbb{P}_{\mathcal{X}'\mathcal{Y}'}$ on $\mathcal{X}'$, where the input space undergoes "type" shifting and the joint distribution has the same label space or different label space (depending to the "type"). We consider a generalized characterization of the open world setting with two types of OOD

$$\mathbf{P}_{\text{wild}} = (1 - \sum_{type} \pi_{type})\mathbb{P}_{\text{in}} + \sum_{type} \pi_{type}\mathbb{P}_{\text{out}}^{type}, \tag{1}$$

where $type = \{\text{semantic}, \text{covariate}\}$, where $\pi_{type}, \sum_{type} \pi_{type} \in (0, 1)$.

**Covariate OOD type:** Taking autonomous driving as an example, a model trained on ID data with sunny weather may experience a covariate shift due to foggy/snowy weather. Under such a covariate shift, a model is expected to generalize to the OOD data—correctly predicting the sample into one of the known classes (e.g., car), despite the shift. $\mathbb{P}_{\text{out}}^{cov}$ is the marginal distribution of covariate shifted data $(\mathcal{X}', \mathcal{Y})$ with distribution $\mathbb{P}_{\mathcal{X}'\mathcal{Y}}$, where the joint distribution has the same label space as the training data, yet the input space undergoes shifting in domain.

**Semantic OOD type:** In autonomous driving example, the model may encounter a semantic shift, where samples are from unknown classes (e.g., bear) that the model has not been exposed to during training. $\mathbb{P}_{\text{out}}^{sem}$ is the marginal distribution when wild data does not belong to any known categories $Y = \{1, 2, ..., K\}$ and therefore should be detected as OOD sample. To detect the semantic OOD data, we train OOD detector $D_\theta(\mathbf{x}, \theta)$ which is a ranking function $g_\theta : \mathcal{X} \mapsto \mathbb{R}$ with parameter $\theta$.

$$D_\theta(\mathbf{x}, \theta) = \left\{ \begin{array}{ll} ID & \text{if } g_\theta(\mathbf{x}) > \lambda \\ \\ OOD & \text{if } g_\theta(\mathbf{x}) \leq \lambda \end{array} \right.$$

The threshold value $\lambda$ is typically chosen so that a high fraction of ID data is correctly classified. This means that the detector $g_\theta$ should predict semantic OOD data as OOD and otherwise predict as ID. An example of $g_\theta$ is energy function $E_\theta(\mathbf{x}) := -\log \sum_{y=1}^K e^{f_\theta^{(y)}(\mathbf{x})}$, where $f_\theta^{(y)}(\mathbf{x})$ denotes the $y$-th element of $f_\theta(\mathbf{x})$, corresponding to label $y$.

**Learning Objectives:** In our setup, we consider the following objective functions:
**ID-Acc** measures the model's performance on $\mathbb{P}_{in}$ which is cross-entropy $\mathbb{E}_{(x,y)\sim\mathbb{P}_{\mathcal{X}\mathcal{Y}}}[\mathcal{L}_{CE}(f(x), y)]$.
**OOD-Acc** measures the OOD generalization ability on $\mathbb{P}_{out}^{cov}$, $\mathbb{E}_{(\mathbf{x},y)\sim\mathbb{P}_{out}^{cov}}[\mathcal{L}_{CE}(f(x), y)]$.
**False positive rate (FPR)** measures the OOD detection $\mathbb{E}_{x\sim\{\mathbb{P}_{out}^{sem}\}}(\mathbb{1}(D_\theta(\mathbf{x}, \theta) = ID))$.

## 2.1 Temp-SCONE Method

In this section, we present our Temp-SCONE methodology that enables performing both OOD generalization and OOD detection in dynamic domains when unlabeled data in the wild is encountered. Our Temp-SCONE method for the first time proposes OWL under temporal shift for streams of wild data which shows superior advantage over the counter part approaches that (1) rely only on the ID data, or (2) address static OOD generalization and OOD detection with strong applications that are deployed in the dynamic open world. In addition, Temp-SCONE maintain SCONE's performance on ID accuracy, covariate shift accuracy, and OOD detection (FPR) on the stream of distinct wild data.
**Leveraging Confidence Score to Enhance both OOD Generalization and Detection:** We define the evolving test-time distribution at time $t$ in (1) as $\mathbf{P}_{\text{wild},t} = (1 - \sum_{type} \pi_{type,t})\mathbb{P}_{\text{in}} + \sum_{type} \pi_{type,t}\mathbb{P}_{\text{out},t}^{type}$, where $type = \{\text{semantic}, \text{covariate}\}$. And $\mathbb{P}_{out,t}$ and $\pi_{type,t}$ may vary over time due to seasonal, contextual factors. Our temporal-SCONE *(Temp-SCONE)* technique, leverages confidence score to enhance OOD detection and generalization with temporal shift Cai et al. [2024], Wu et al. [2025], Chang et al. [2025].
**Definition:***(ATC Garg and Balakrishnan [2022])* Consider softmax prediction of the function $f$, and two such score functions: $s(f_\theta(x)) = \max_{j\in\mathcal{Y}} f_j(x)$ (maximum confidence) and $s(f_\theta(x)) = \sum_j f_j(x) \log f_j(x)$ (negative entropy). ATC identifies a threshold and the error estimate is given by the expected number of points that obtain a score less than $\delta$ i.e.

$$ATC(s) := \mathbb{E}_{\mathbb{P}_{in}}\left[\mathbb{1}\{s(f_\theta(x)) < \delta\}\right]. \tag{2}$$

In this paper, we propose a temporal shift accurance based on ATC criteria (2). We also use average confidence $AC(s) := \mathbb{E}_{\mathbb{P}_{in}}[\mathbb{1}\{s(f_\theta(x))\}]$ as secondary confidence score to compare against ATC. **Definition:** *(Temporal Shift)* Consider marginal distribution of the labeled data $(\mathcal{X}_t, \mathcal{Y}_t)$ at time step $t$ ($\mathbb{P}_{in}^t$). We define *temporal shift* for the classifier $f_\theta(x)$ iff the ATC is shifted over time.

$$\left| \mathbb{E}_{\mathbb{P}_{in}^{t+1}}\left[\mathbb{1}\{s(f(x)) < \delta\}\right] - \mathbb{E}_{\mathbb{P}_{in}^t}\left[\mathbb{1}\{s(f(x)) < \delta\}\right] \right| \leq \epsilon, \tag{3}$$

where $\epsilon \geq 0$ is small constant. Note that the classifier is trained on an online dataset.

*Temp-SCONE objective function:* Given access to wild samples $\{\tilde{\mathbf{x}}_{1t}, \ldots, \tilde{\mathbf{x}}_{mt}\}$ from wild data with distribution $\mathbb{P}_{\text{wild},t}$ along with labeled ID samples $(\mathbf{x}_1, y_1), ..., (\mathbf{x}_n, y_n)$. Denote the combination of covariate shifted $\{\mathbf{x}^c_{1t}, \ldots, \mathbf{x}^c_{m_{ct}}\}$ and ID data $\{\mathbf{x}_{1t}, \ldots, \mathbf{x}_{m_{id_t}}\}$ by $\{\mathbf{x}^{id,c}_{1t}, \ldots, \mathbf{x}^{id,c}_{m_{id,ct}}\}$. Here $mt = m_{ct} + m_{st} + m_{id_t}$ are the size of covariate shifted, semantic shifted, and ID sample sizes.

$$\text{Temp-SCONE Optimization with ATC} \rightarrow \arg\min_\theta \frac{1}{m} \sum_{i=1}^m \mathbb{1}\{\mathbb{E}_\theta(\tilde{\mathbf{x}}_{it}) \leq 0\}$$

$$\text{s. t. } \frac{1}{n}\sum_{j=1}^n \mathbb{1}\{\mathbb{E}_\theta(\mathbf{x}_{jt}) \geq \eta\} \leq \alpha, \quad \frac{1}{n}\sum_{j=1}^n \mathbb{1}\{\hat{y}(f_\theta(\mathbf{x}_{jt}) \neq y_j\} \leq \tau,$$

$$\left| \frac{1}{m_{ct}}\sum_{r=1}^{m_{ct}} \mathbb{1}\{s(f_\theta(\mathbf{x}^{id,c}_{rt})) < \delta\} - \frac{1}{m_{c(t-1)}}\sum_{r=1}^{m_{c(t-1)}} \mathbb{1}\{s(f_\theta(\mathbf{x}^{id,c}_{r(t-1)})) < \delta\} \right| \leq \epsilon. \tag{4}$$

In (4), $m_{id,ct} = m_{ct} + m_{id_t}$ and the ATC and AC are computed on $\{\mathbf{x}^{id,c}_{1t}, \ldots, \mathbf{x}^{id,c}_{m_{id,ct}}\}$. And the energy function $E_\theta(x)$ is defined by $E_\theta(x) = -\log \sum_{y=1}^K e^{f_\theta^{(y)}(x)}$, where $f_\theta^{(y)}(x)$ is the logit value for class $y$. Our Temp-SCONE objective function relies on WOODs Katz-Samuels et al. [2022] and SCONE Bai et al. [2023], and enforces the ID data to have energy smaller than the margin $\eta$ (a negative value), a margin controller for OOD decision boundary with respect to the ID data, while optimizing for the level-set estimation based on the energy function. The temporal loss (the last line in (4)) controls the confidence level (ATC) turbulence of both ID and covariate shifted datasets through dynamic domains.

---

**Algorithm 1** Differentiable Temporal Loss with Mode Switching and Adaptive Weighting

---

**Input:** In-dist. data $\mathcal{D}_{in}^t$, covariate OOD data $\mathcal{D}_{cov}^t$, model $f_\theta$ at timestep $t$
**Input:** State store state with previous scores, mode $\in \{\texttt{ATC}, \texttt{AC}\}$, smoothing $\omega$, base weight $\lambda_{\text{base}}$, max drift $\Delta_{\max}$
**Output:** Temporal loss $\mathcal{L}_{\text{temp},t}(f_\theta)$
**if** $t = 0$ **then**
 | $\mathcal{L}_{\text{temp},t} \leftarrow 0$ // initialize (grad-enabled zero in implementation) **return** $\mathcal{L}_{\text{temp},t}$
**end**
// Differentiable confidence/ATC scores at timestep $t$
**if** $mode = \texttt{ATC}$ **then** $s_{in}^t \leftarrow \text{DiffATC}(f_\theta, \mathcal{D}_{in}^t; \delta = \Delta_{\max}, \omega)$   $s_{cov}^t \leftarrow \text{DiffATC}(f_\theta, \mathcal{D}_{cov}^t; \delta = \Delta_{\max}, \omega)$
**else if** $mode = \texttt{AC}$ **then** $s_{in}^t \leftarrow \text{DiffAC}(f_\theta, \mathcal{D}_{in}^t)$   $s_{cov}^t \leftarrow \text{DiffAC}(f_\theta, \mathcal{D}_{cov}^t)$
// Fetch previous-time scores from state
$p_{in}^{t-1} \leftarrow$ state[last in-score for mode]   $p_{cov}^{t-1} \leftarrow$ state[last cov-score for mode]
// Asymmetric temporal drift (penalize ID decreases and COV increases)
$d_{\text{id}} \leftarrow \left[p_{in}^{t-1} - s_{in}^t\right]_+$   $d_{\text{cov}} \leftarrow \left[s_{cov}^t - p_{cov}^{t-1}\right]_+$   $d_{\text{tot}} \leftarrow d_{\text{id}} + d_{\text{cov}}$
// Adaptive temporal weighting
$w_{\text{temp}} \leftarrow \text{AdaptiveWeight}(d_{\text{id}}, d_{\text{cov}}; \lambda_{\text{base}}, \Delta_{\max})$
// Final temporal loss
$\mathcal{L}_{\text{temp},t}(f_\theta) \leftarrow w_{\text{temp}} \cdot d_{\text{tot}}$
// Update state (e.g., append loss, weight, drift; optionally log)
state $\leftarrow \text{UpdateState}(\text{state}, \mathcal{L}_{\text{temp},t}, w_{\text{temp}}, d_{\text{id}}, d_{\text{cov}}, t)$
**return** $\mathcal{L}_{\text{temp},t}(f_\theta)$

---

*How to train Temp-SCONE model?:* To demonstrate our Temp-SCONE method, we employed the SCONE approach and executed three main steps: *(Step 1)* load wild data $\mathcal{D}_{aux}^t$ that is combination

of ID, covariate and semantic shifted data, $\mathcal{D}_{in}^t, \mathcal{D}_{out}^{cov,t}, \mathcal{D}_{out}^{sem,t}$; *(Step 2)* compute loss functions $\mathcal{L}_{CE}^t, \mathcal{L}_{in}^t, \mathcal{L}_{out}^t$ and $\mathcal{L}_{temp}^t$; *(Step 3)* backpropagate and update parameter $\theta$ based on loss function $\mathcal{L}_{\text{total}}^t = \mathcal{L}_{CE}^t + \lambda_{\text{out}} \cdot \mathcal{L}_{\text{out}}^t + \lambda_{temp}\mathcal{L}_{temp}^t$, where $\lambda_{\text{out}}$ and $\lambda_{temp}$ are hyperparameters. $\mathcal{L}_{\text{total}}^t$ is the loss function that aligns with Temp-SCONE objective function (4). The 0/1 loss is not differentiable, hence, we will replace it with a smooth approximation given by the binary sigmoid loss function. the algorithms 1 and 2, illustrates the details of steps above.

**Algorithm 1 notations:** $\mathcal{D}_{in}^t$ is ID data $\{\mathbf{x}_{1t}, \ldots, \mathbf{x}_{m_{id_t}}\}$, $\mathcal{D}_{cov}^t$ is covariate-shifted OOD data $\{\mathbf{x}^c_{1t}, \ldots, \mathbf{x}^c_{m_{ct}}\}$. $f_\theta$ is the classifier with parameters $\theta$. $s_{in}^t, s_{cov}^t$ denote differentiable ATC (or AC) scores on $\mathcal{D}_{in}^t$ and $\mathcal{D}_{cov}^t$, respectively. $p_{in}^{t-1}, p_{cov}^{t-1}$ are the corresponding scores stored from timestep $t - 1$. The temporal drifts are $d_{\text{id}} = [p_{in}^{t-1} - s_{\text{in}}^t]_+$ (ID confidence decrease) and $d_{\text{cov}} = [s_{\text{cov}}^t - p_{\text{cov}}^{t-1}]_+$ (COV confidence increase). $w_{\text{temp}}$ is the adaptive temporal weight based on $d_{\text{id}}, d_{\text{cov}}$. The temporal loss is $\mathcal{L}_{\text{temp},t}(f_\theta) = w_{\text{temp}} \cdot (d_{\text{id}} + d_{\text{cov}})$.

**Algorithm 2 notations:** $\mathcal{D}_{out}^{Sem,t}$ denotes semantic OOD data and $\{D^t\}_{t=0}^T$ denotes wild data. $\tilde{x}_{\text{aux}}^t$ is batch of wild data $\{D^t\}_{t=0}^T$. $y_{in}^t$ is the label of ID data. $z^t$ is the logit layer of the classifier $f_\theta$ and ID energy, $E_{in}^t$ and OOD energy, $E_{out}^t$ are computed from $z_{cls}^t$ and $z_{aux}^t$, respectively.

---

**Algorithm 2** Training Temp-SCONE

**Input:** $\{D^t\}_{t=0}^T$ (A combination of $\mathcal{D}_{\text{id}}^t$, $\mathcal{D}_{out}^{cov,t}$, and $\mathcal{D}_{out}^{Sem,t}$ datasets), Model $f_\theta$, logistic layer $g_\theta$ for energy-based detection, hyperparameters $\eta, \lambda_{\text{in}}, \lambda_{out}, \lambda_{temp}, \text{FPR}_{\text{cutoff}}, \delta, \text{lr}_\lambda, \texttt{ce\_tol}$, and penalty multipliers $\lambda, \lambda_2$

**Output:** Trained OOD detector and generalized model $f_\theta$

**for** *t = 0* **to** *T* **do**

  Load $D_{\text{in}}^t, D_{out}^{cov,t}, D_{out}^{sem,t}$

  Compute baseline classification loss $\leftarrow \mathcal{L}(f_\theta)$ loss on $D_{\text{in}}^t$

  **for** *epoch = 1* **to** *E* **do**

    // -- Compute Temporal Loss from Algorithm 1 --

    // -- Mini-batch Training Loop --

    **foreach** *mini-batch* $(x_{in}^t, y_{in}^t), x_{out}^{cov,t}, x_{out}^{sem,t}$ **do**

      $\tilde{x}_{\text{aux}}^t \leftarrow \texttt{MixBatches}(x_{in}^t, x_{out}^{cov,t}, x_{out}^{sem,t})$ $x^t \leftarrow \texttt{concat}(x_{in}^t, \tilde{x}_{\text{aux}}^t), y^t \leftarrow y_{in}^t$

      $z^t = f_\theta(x^t), z_{\text{cls}}^t = z[: |x_{\text{in}}^t|]$, and $z_{\text{aux}}^t = z[: |\tilde{x}_{\text{aux}}^t|]$ $\mathcal{L}_{\text{CE}}(f_\theta) \leftarrow \text{CrossEntropy}(z_{\text{cls}}^t, y^t)$

      // -- Energy-based OOD losses --

      $E_{\text{in}}^t \leftarrow \text{logsumexp}(z_{\text{cls}}^t), E_{\text{out}}^t \leftarrow \text{logsumexp}(z_{\text{aux}}^t)$ $\mathcal{L}_{\text{in}}^t = \text{sigmoid}(g_\theta(E_{\text{in}}^t))$ $\mathcal{L}_{\text{out}}^t = \text{sigmoid}(-g_\theta(E_{\text{out}}^t - \eta))$

      // -- Augmented Lagrangian Terms --

      $\texttt{in\_constraint} \leftarrow \mathcal{L}_{\text{in}}^t - \text{FPR}_{\text{cutoff}}$ $\texttt{alm}_{\text{in}} \leftarrow \lambda \cdot \texttt{in\_constraint} + \frac{\lambda_{\text{in}}}{2} \cdot (\texttt{in\_constraint})^2$

      $\mathcal{L}_{\text{total}}^t \leftarrow \mathcal{L}_{CE}^t + \lambda_{\text{out}} \cdot \mathcal{L}_{\text{out}}^t + \texttt{alm}_{\text{in}} + \lambda_{temp}\mathcal{L}_{temp}^t$

      Backpropagate and update model parameters $\theta$

    **end**

    // -- Lagrange Multiplier Updates --

    Compute $\mathcal{L}_{\text{in}}^t$ and $\mathcal{L}_{\text{CE}}^t$ over $D_{\text{in}}^t$ $\lambda \leftarrow \lambda + lr_\lambda \cdot (\mathcal{L}_{\text{in}}^t - \text{FPR}_{\text{cutoff}})$ $\lambda_2 \leftarrow \lambda_2 + lr_\lambda \cdot (\mathcal{L}_{\text{CE}}^t - \texttt{ce\_tol} \cdot \mathcal{L}(f_\theta))$

  **end**

**end**

---

**Differences between SCONE and Temp-SCONE:** The SCONE framework builds on WOODS Katz-Samuels et al. [2022] by introducing an energy margin $\eta < 0$ to separate ID and covariate-shifted samples from semantic OOD. Specifically, SCONE leverages the energy function $E_\theta(x)$, which assigns negative energy to ID data and positive energy to OOD data. In WOODS, the boundary $E_\theta(x) = 0$ often misclassifies covariate-shifted samples as semantic OOD; SCONE resolves this by requiring $E_\theta(x) < \eta$, which (1) pushes ID deeper into the negative region and (2) pulls covariate-shifted samples below the margin. Thus, everything to the left of $\eta$ is ID/covariate-OOD (semantically valid), and everything to the right of 0 is semantic OOD. Temp-SCONE leverages the same mechanism but further addresses *temporal shifts and average confidence control over time*, which SCONE

does not consider. It introduces a *temporal loss* that regularizes fluctuations in confidence across sequential domains. Using differentiable ATC/AC, Temp-SCONE tracks the stability of model confidence, penalizing drifts beyond a tolerance $\epsilon$ with an *adaptive temporal weighting scheme* that applies stronger correction when drift is large. This prevents "confidence turbulence" during domain transitions and helps maintain reliable decision boundaries. In summary, SCONE enforces a static energy margin to separate ID/covariate vs. semantic OOD, while Temp-SCONE augments this with a *time-aware consistency mechanism* that stabilizes the decision rule under evolving distributions.

## 3    Experiments

**Datasets and Experimental Setup** We evaluate the effectiveness of Temp-SCONE across diverse datasets and model architectures, focusing on robustness under two key settings: (1) Dynamic (temporal) datasets that evolve gradually over time, and (2) Distinct datasets with no temporal continuity, representing strong domain shifts. Each timestep includes an ID dataset, its covariate-shifted (corrupted) variant, and a semantic OOD dataset.

**Dynamic Datasets.** We use the CLEAR dataset Lin et al. [2021], which spans 10 temporal stages, each representing a distinct time period. For every timestep, we define ID (original data), covariate-shifted (Gaussian noise), and semantic OOD (Places365 Zhou et al. [2017]) splits to evaluate OOD detection and generalization under temporal drift. As a complementary benchmark, we use the Yearbook dataset Ginosar et al. [2015], containing grayscale portraits of U.S. high school students over a century, divided into 7 temporal stages with 11 balanced classes per stage. Similar to CLEAR, we apply Gaussian noise for covariate shifts, while FairFace Kärkkäinen and Joo [2019] serves as the semantic OOD dataset, introducing demographic and contextual diversity.

**Distinct Datasets.** For the distinct (non-temporal) setting, we conduct experiments using four different ID datasets and varying semantic OOD datasets: ID for timesteps 1-4: are CIFAR-10 Krizhevsky [2009] $\rightarrow$ Imagenette Howard [2019] $\rightarrow$ CINIC-10 Darlow et al. [2018] $\rightarrow$ STL-10 Coates et al. [2011] as the ID datasets, each with its own covariate-shifted versions generated using Gaussian noise and Defocus blur corruptions. In the distinct experiment, the semantic OOD dataset changes with the timestep: timestep 1 uses LSUN-C Yu et al. [2015], timestep 2 uses SVHN Netzer et al. [2011], timestep 3 uses Places365 Zhou et al. [2017], and timestep 4 uses DTD Cimpoi et al. [2014] (Textures). We perform three additional experiments that are provided in Appendix by applying semantic OOD dataset LSUN-C, SVHN, or Places365) across all timesteps.

**Training Procedure.** In both dynamic and distinct settings, the model is first trained on ID data from timestep 1 using a standard classification objective, serving as initialization for the Temp-SCONE framework. In the dynamic setting, the model is then trained sequentially from timestep 1 to timestep 10 on the CLEAR dataset, where each timestep represents a distinct temporal distribution. In the distinct setting, the same initialized model is fine-tuned independently on each dataset (CIFAR-10, Imagenette, CINIC-10, STL-10), treating them as separate domains.

**Model Architectures and Optimization.** We evaluate TEMP-SCONE using two backbone architectures: a convolutional neural network WideResNet-40-2 (WRN) Debgupta et al. [2020] and vision transformer ViT (DeiT-Small) Han et al. [2022]. All models are trained using stochastic gradient descent (SGD) with Nesterov momentum of 0.9, a weight decay of 0.0005, and a batch size of 128. In the dynamic setting (CLEAR), we use a multi-step learning rate schedule, starting at 0.0001 and decaying by a factor of 0.5 at 50%, 75%, and 90% of training. In the distinct setting, we also use an initial learning rate of 0.0001 for timestep 1 (CIFAR 10), and multiply it by a factor of 5 for timestep 2 (Imagenette), timestep 3 (CINIC-10) and timestep 4 (STl-10) to account for their increased visual complexity.

**Temporal Regularization.** We integrated the TEMP-SCONE framework in two variants, each using a different metric for temporal consistency. One variant uses ATC (2) to measure and regularize the change in confidence between timesteps, while the other variant uses AC for the same purpose. In both cases, we apply a temporal loss term if the chosen metric's drift exceeds a defined threshold, helping the model maintain stable confidence across shifts. We ran experiments with both ATC-based and AC-based TEMP-SCONE variants to evaluate their effectiveness in reducing OOD detection errors and maintaining performance over time.

In our results, we report ID Acc. which is the accuracy on the clean ID test set, OOD Acc. which is the accuracy on the Gaussian-corrupted version of the test set, and finally FPR95, which is false positive rate when 95 percent of ID examples are correctly classified.

# 4 Results and Discussion

**Temp-SCONE outperforms SCONE on dynamic domains.** We present experiments on CLEAR and Yearbook to show that Temp-SCONE consistently outperforms SCONE across all timesteps. In Fig. 1, Yearbook serves as ID and FairFace as OOD for both WRN and ViT. Results highlight Temp-SCONE's stability benefits, with superior ID accuracy (left), OOD accuracy (middle), and lower FPR95 (right). For ViT, SCONE exhibits volatility—early drops in ID/corrupted accuracy and high FPR95—while Temp-SCONE with AC/ATC yields smoother trajectories, higher accuracies, and lower FPR95 in early/mid timesteps, reducing forgetting and improving robustness under appearance drift. All methods show a U-shape over time, but Temp-SCONE, especially ATC, drives stronger recovery in ID/corrupted accuracy, while AC provides steadier OOD detection and ATC trades stability for more aggressive adaptation. Across both backbones, at least one Temp-SCONE variant (AC or ATC) dominates SCONE on the primary robustness axis—accuracy under corruption—while also improving temporal stability of ID performance and delivering competitive or better OOD calibration on ViT. Thus, under temporal drift, Temp-SCONE with AC or ATC offers a strictly stronger robustness profile than SCONE.

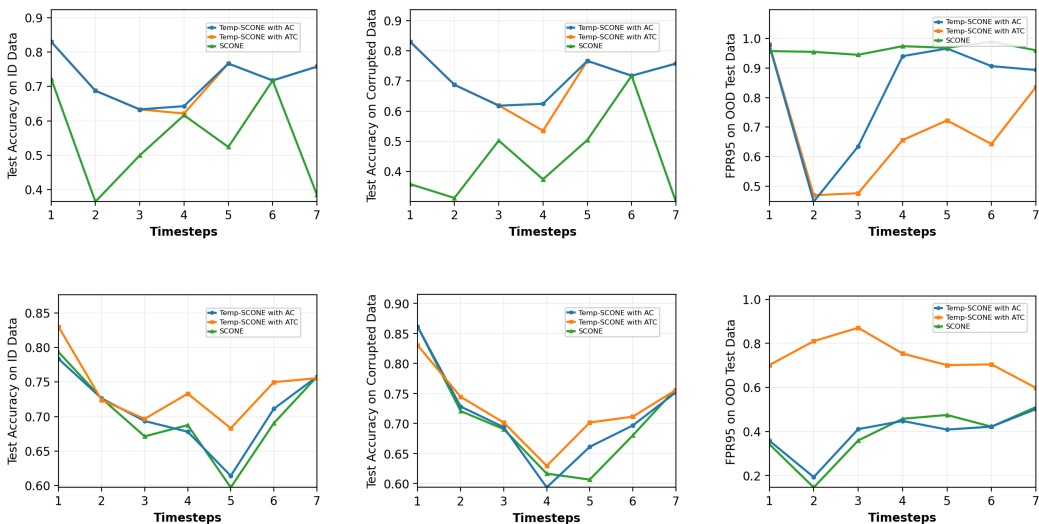

Figure 1: Dynamic Data (YearBook - 7 timesteps), FairFace is OOD data, (top row WRN, bottom row ViT). Columns show ID Acc.↑, OOD Acc.↑, FPR95 ↓.

Our second set of experiments treats CLEAR as the ID benchmark and evaluates OOD detection against Places365 for both WRN and ViT across ten timesteps. As shown in Fig. 2, adding the temporal stability term (Temp-SCONE with AC/ATC) consistently improves robustness under distribution shift. On WRN, Temp-SCONE shows slightly lower early ID accuracy than SCONE but achieves larger and more persistent gains on corrupted data, demonstrating stronger robustness to covariate shift. SCONE's early advantage reflects mild overfitting to the initial domain, while Temp-SCONE's temporal regularization mitigates this, yielding smoother performance and better generalization over time. Although OOD detection is mixed, Temp-SCONE maintains higher corrupted accuracy and comparable late-stage FPR95, offering a more balanced robustness profile. The non-monotonic trends align with typical temporal drift behavior, where early adaptation and later shifts jointly shape performance. On ViT, Temp-SCONE strictly dominates: both AC and ATC achieve higher corrupted and clean accuracy than SCONE, and ATC attains the lowest FPR95 in later timesteps, indicating improved OOD calibration where drift accumulates. Overall, CLEAR results confirm that introducing temporal consistency yields a Pareto improvement on ViT and a clear robustness win on WRN, establishing Temp-SCONE (AC/ATC) as preferable to SCONE for dynamic data.

**Temp-SCONE maintains SCONE's performance on distinct data.** Fig. 3 shows results on four distinct datasets (CIFAR-10, Imagenette, CINIC-10, STL-10), where each timestep corresponds to a different dataset. Across both WRN and ViT, SCONE and Temp-SCONE curves overlap, indicating no advantage from temporal regularization when domains lack continuity. The AC and ATC

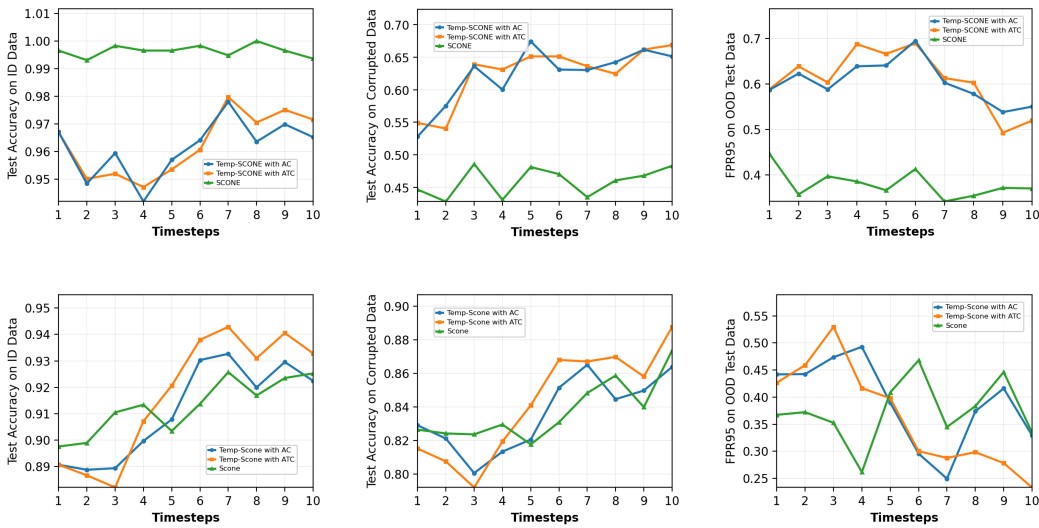

Figure 2: Dynamic Data (CLEAR - 10 timesteps), Places365 is OOD data, (top row WRN, bottom row ViT). Columns show ID Acc.↑, OOD Acc.↑, FPR95 ↓.

variants of Temp-SCONE behave almost identically, further confirming that temporal consistency provides no advantage in this setting. A consistent trend emerges: SCONE is not robust to distinct datasets—FPR95 rises sharply after the first timestep, while ID and OOD accuracy drop, especially for ViTs. Temp-SCONE inherits this limitation, as its temporal loss assumes gradual drift and cannot handle fully disjoint shifts. While WRNs retain slightly better stability, both backbones collapse under distinct domains.

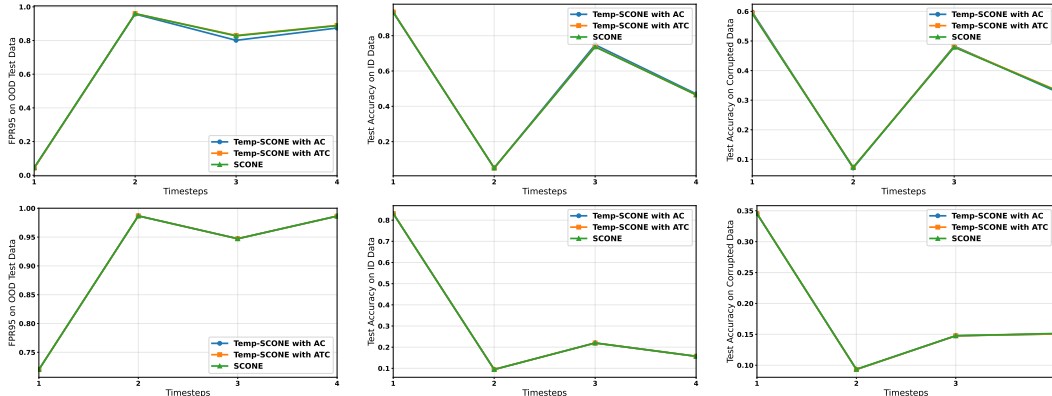

Figure 3: Distinct Data - CIFAR-10 → Imagenette → CINIC-10 → STL-10 are four ID timesteps. Semantic OOD dataset changes with the timestep: timestep 1 uses LSUN-C, timestep 2 uses SVHN, timestep 3 uses Places365, and timestep 4 uses DTD (Textures), (top row WRN, bottom row ViT). Columns show ID Acc.↑, OOD Acc.↑, FPR95 ↓.

## 5 Theoretical Insights

Motivated by the success of WOODs Katz-Samuels et al. [2022], SCONE Bai et al. [2023], and inspired by theoretical investigations in Zhang et al. [2024], Tong et al. [2021], we have studied generalization error $(GErr_{t+1}(f))$ of model $f_\theta$ for two time steps $t$ and $t+1$. We assume: **[A1]** At time step $t$, $TV(p(y_t|x_t)\|\mathcal{U})$ is constant. **[A2]** At time step $t$, $F_f^{\theta_1}$ The class distributions predicted by $f$ and $p^{\theta_2}(y_t|x_t)$ have same distribution with different parameter $\theta_1$ and $\theta_2$, respectively and

$\theta_1 - \theta_2 = \delta$, where $\delta$ is bounded. **[A3]** There exist a constant (say $Z_t$), s.t.

$$\mathbb{E}_{\mathbb{P}_{out}^{t+1,cov}} H(p(y_{t+1}|x_{t+1})) - \mathbb{E}_{\mathbb{P}_{out}^{t,cov}} H(p(y_t|x_t))) \geq Z_t + Conf_t - Conf_{t+1}.$$

**Theorem 5.1.** *Let $\mathbb{P}^{t,cov}$ and $\mathbb{P}_{test}^{t,sem}$ be the covariate-shifted OOD and semantic OOD distribution. Denote $GErr_{t+1}(f)$ the generalization error at time $t$. Let $L_{reg}$ be the OOD detection loss devised for MSP detectors Hendrycks et al. [2018], i.e., cross-entropy between predicted distribution $f_\theta$ and uniform distribution. Then at two time steps $t$ and $t+1$ and under assumptions* **[A1]**-**[A3]***, we have*

$$GErr_{t+1}(f) - GErr_t(f) \geq -\tilde{\kappa}\, \Delta_{t \to t+1}^{cov,sem} - \tilde{\kappa}\, \Xi_{t \to t+1}^{sem} - \overline{\delta}_t^2 \, \mathbb{E}_{\mathbb{P}_{out}^{t,cov}} (I_F(\theta))$$
$$+ C_{t \to t+1} + Conf_t - Conf_{t+1}, \tag{5}$$

$$where \quad \Delta_{t \to t+1}^{cov,sem} := d_{\mathcal{F}}(\mathbb{P}_{out}^{t+1,cov}, \mathbb{P}_{out}^{t+1,sem}) + d_{\mathcal{F}}(\mathbb{P}_{out}^{t,cov}, \mathbb{P}_{out}^{t,sem})$$

$$and \quad \Xi_{t \to t+1}^{sem} := \mathbb{E}_{\mathbb{P}_{out}^{t+1,sem}} \sqrt{\frac{1}{2}(\mathcal{L}_{reg}(f) - \log K)} + \mathbb{E}_{\mathbb{P}_{out}^{t,sem}} \sqrt{\frac{1}{2}(\mathcal{L}_{reg}(f) - \log K)}.$$

*And $C_{t \to t+1} = C_{t+1} - C_t + B_t + Z_t$ and $\delta_t$ are constants and $\overline{\delta}_t^2 = \frac{\log e}{2}\delta_t^2$. Here $d_{\mathcal{F}}(\mathbb{P}_{out}^{t,cov}, \mathbb{P}_{out}^{t,sem})$ is disparity discrepancy with total variation distance) (TVD) that measures the dissimilarity of covariate-shifted OOD and semantic OOD. $Conf$ is maximum confidence $Conf(f_\theta) := \max_{j \in \mathcal{Y}} f_j(x)$, and $I_f(\theta)$ is Fisher information Cramér [1999].*

The details and proof are deferred in Appendix. Our theoretical finding demonstrates that for MSP detectors (without any OOD detection regularization), at two timesteps $t$ and $t+1$, the OOD detection objective difference conflicts with OOD generalization difference. In addition, the generalization error difference over time is not only negatively correlated with OOD detection loss that the model minimizes, it also negatively correlated to the Fisher information of the network parameter under $\mathbb{P}_{out}^{t,cov}$. The OOD generalization error at $t+1$ and $t$ is positively correlated with confidence difference over the same period. It is important to mention that similar to Zhang et al. [2024] our theorem is applicable for all MSP-based OOD detectors. The inherent motivation of OOD detection methods lies in minimizing the OOD detection loss in $\mathbb{P}_{out}^{t,sem}$ under test data, regardless of the training strategies used.

# 6 Related Work

**Robustness for Wild Data.** Recent work has addressed OOD detection and generalization in open-world settings. SCONE enhances robustness to "wild" data comprising ID, covariate-shifted, and semantic-shifted samples by imposing margin-based constraints that separate semantic OOD while keeping covariate OOD aligned with ID Bai et al. [2023]. Beyond fully automated approaches, human-assisted frameworks have also been explored: AHA leverages selective annotation in the maximum disambiguation region to better separate covariate and semantic shifts and has been shown to outperform SCONE in wild-data settings Bai et al. [2024]. **OOD Detection in Time-Series.** Most OOD detection methods are developed for vision and language, with limited assessment in time-series. A recent study provides a comprehensive analysis of modality-agnostic OOD algorithms on multivariate time-series, showing that many SOTA methods transfer poorly, while deep feature–based approaches appear more promising Gungor et al. [2025]. This complements our focus: rather than benchmarking generic methods on time-series, we target wild OOD classification with temporal dynamics, where distributions evolve across time. **Temporal OOD Detection.** Recent work addresses OOD detection under temporally evolving settings via sliding-window calibration, temporal consistency or ensembling, and test-time/continual adaptation Wang et al. [2020], Sun et al. [2020], Gao et al. [2023], Wu et al. [2023]. These approaches stabilize predictions but largely treat OOD dynamics in aggregate, without explicitly disentangling covariate vs. semantic OOD or providing fine-grained stability across timesteps. Complementarily, Temp-SCONE introduces a confidence-driven temporal regularization that leverages ATC (and AC) to penalize confidence turbulence between domains while retaining SCONE's energy-margin separation for robust covariate and semantic OOD detection.

# 7  Conclusion

In this work, we introduced Temp-SCONE, a temporally-consistent extension of SCONE that addresses the challenges of OOD detection and generalization under evolving data distributions. By integrating confidence-based metrics with a temporal regularization loss, Temp-SCONE stabilizes decision boundaries across timesteps and mitigates confidence turbulence during domain transitions. Our experimental results on both dynamic datasets and distinct datasets highlight several key findings: (1) Temp-SCONE, significantly improves robustness and OOD calibration in temporally evolving domains, particularly under covariate shifts under either WRN or ViT network; (2) on distinct datasets with abrupt domain changes, Temp-SCONE maintains parity with SCONE, underscoring the limits of temporal regularization when no temporal continuity exists; and (3) vision transformers benefit most from temporal consistency, demonstrating reduced instability and improved reliability under drift.

## Acknowledgments

This work has been partially supported (Aditi Naiknaware and Salimeh Yasaei Sekeh) by NSF CAREER CCF-2451457, and (Sanchit Sigh and Hajar Homayouni) has benefited from the Microsoft Accelerating Foundation Models Research (AFMR) grant program. The findings are those of the authors only and do not represent any position of these funding bodies.

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

# 8 Appendix

## 8.1 Theoretical Proofs

**Lemma .1.** *At time steps $t$ and $t+1$, if $H(p(y_t|x_t)) \leq H(p(y_{t+1}|x_{t+1}))$ then*

$$Conf_t = \max_{y_t \in \mathcal{Y}_t} p(y_t|x_t) \geq \max_{y_{t+1} \in \mathcal{Y}_{t+1}} p(y_{t+1}|x_{t+1}) = Conf_{t+1}.$$

**Proof:** For $K$ classes at both time $t$ and $t+1$, denote $p_t^* := \max_{y_t \in \mathcal{Y}_t} p(y_t|x_t)$ and $p_{t+1}^* := \max_{y_{t+1} \in \mathcal{Y}_{t+1}} p(y_{t+1}|x_{t+1})$. Suppose $p_t^* = P(y_t = k_1|x_t)$ and $p_{t+1}^* = P(y_t = k_2|x_{t+1})$. Now set $p_t = (p_t^*, 1 - p_t^*)$, where $1 - p_t^*$ is split among classes $\{1, \ldots, K\}/k_1$ and $1 - p_{t+1}^*$ is split among classes $\{1, \ldots, K\}/k_2$. This approximates the entropy as

$$H(p_t) = -p_t^* \log p_t^* - \sum_{i \in \{1, \ldots, K\}/k_1} p_{it} \log p_{it}, \tag{6}$$

where $p_{it} = \frac{1 - p_t^*}{K - 1}$. And (6) is simplified as

$$H(p_t) = -p_t^* \log p_t^* - (1 - p_t^*) \log \frac{1 - p_t^*}{K - 1}. \tag{7}$$

Equivalently

$$H(p_{t+1}) = -p_{t+1}^* \log p_{t+1}^* - (1 - p_{t+1}^*) \log \frac{1 - p_{t+1}^*}{K - 1}. \tag{8}$$

Because $H(p_t) \leq H(p_{t+1})$ and from (7) and (8), we implies that $p_t^* \geq p_{t+1}^*$.

**Lemma .2.** *(Theorem 1, (Zhang et al. [2024])) The generalization error at time step $t$, $GErr_t$, is standard cross entropy loss for hypothesis $f \in \mathcal{F}$ under covariant shift $\mathbb{P}^{cov}$. $GErr_t$ is lower bounded by*

$$GErr_t(f) \geq -\frac{1}{2\kappa} \mathbb{E}_{\mathbb{P}_{out}^{t,sem}} \sqrt{\frac{1}{2}(\mathcal{L}_{reg}(f) - \log K)} \tag{9}$$

$$-\frac{1}{2\kappa} d_{\mathcal{F}}(\mathbb{P}_{out}^{t.cov}, \mathbb{P}_{out}^{t,sem}) + C_t + \mathbb{E}_{\mathbb{P}_{out}^{t,cov}} H(p(y_t|x_t)),$$

,

*where $C_t$ is constant.*

**Lemma .3.** *(Lemma 1, (Zhang et al. [2024])) For any $f \in \mathcal{F}$, we have*

$$\mathbb{E}_{\mathbb{P}_{out}^{t,cov}} TV(F_f \| \mathcal{U}) \leq \mathbb{E}_{\mathbb{P}_{out}^{t,sem}} TV(F_f \| \mathcal{U}) \tag{10}$$

$$+ d_{\mathcal{F}}(\mathbb{P}_{out}^{t.cov}, \mathbb{P}_{out}^{t,sem}) + \lambda,$$

*where $\lambda$ is a constant independent of $f$. $\mathcal{U}$ is the $K$-classes uniform distribution. $\mathbb{P}_{out}^{t,cov}$ is the covariate-shifted OOD distribution at time $t$. $\mathbb{P}_{out}^{t,sem}$) is the semantic OOD distribution at time $t$.*

**Lemma .4.** *(Lemma 3, (Zhang et al. [2024])) Denote the OOD detection loss used for MSP detectors as $\mathcal{L}_{reg}$, then we have*

$$\mathbb{E}_{\mathbb{P}_{out}^{t,sem}} \left( TV(F_f \| \mathcal{U}) \right) \leq \mathbb{E}_{\mathbb{P}_{out}^{t,sem}} \sqrt{\frac{1}{2}(\mathcal{L}_{reg}(f) - \log K)}. \tag{11}$$

**Lemma .5.** *The generalization error at time step $t$, $GErr_t$, is standard cross entropy loss for hypothesis $f \in \mathcal{F}$ under covariant shift $\mathbb{P}^{cov}$. $GErr_t$ is lower bounded by*

$$GErr_t(f) \leq \frac{loge}{2} \mathbb{E}_{\mathbb{P}_{out}^{t,sem}} \sqrt{\frac{1}{2}(\mathcal{L}_{reg}(f) - \log K)} + \frac{loge}{2} d_{\mathcal{F}}(\mathbb{P}_{out}^{t.cov}, \mathbb{P}_{out}^{t,sem}) \tag{12}$$

$$+ C_t + \frac{loge}{2} \mathbb{E}_{\mathbb{P}_{out}^{t,cov}} \left( \mathcal{X}^2(p(y_t|x_t) \| F_f(x_t)) \right) + H(p(y_t|x_t)), \tag{13}$$

*where $C_t$ is constant.*

**Proof:**

$$\begin{aligned}
GErr_t(f) &:= \mathbb{E}_{\mathbb{P}_{out}^{t,cov}} \mathcal{L}_{CE}(f(x_t, y_t)) \\
&= \mathbb{E}_{\mathbb{P}_{out}^{t,cov}} KL(p(y_t|x_t)\|F_f(x_t)) + H(p(y_t|x_t))
\end{aligned}$$

$$\leq \frac{loge}{2}\mathbb{E}_{\mathbb{P}_{out}^{t,cov}}\left(TV(p(y_t|x_t)\|F_f(x_t)) + \mathcal{X}^2(p(y_t|x_t)\|F_f(x_t))\right) + H(p(y_t|x_t)) \quad (14)$$

$$\leq \frac{loge}{2}\mathbb{E}_{\mathbb{P}_{out}^{t,cov}}\left(TV(p(y_t|x_t)\|\mathcal{U}) + TV(F_f(x_t)\|\mathcal{U})\right) \quad (15)$$

$$+ \mathbb{E}_{\mathbb{P}_{out}^{t,cov}}\left(\mathcal{X}^2(p(y_t|x_t)\|F_f(x_t))\right) + \mathbb{E}_{\mathbb{P}_{out}^{t,cov}}H(p(y_t|x_t)) \quad (16)$$

*where from Sason and Verdú [2016], we have*

$$\mathcal{X}^2(P\|Q) + 1 = \int \frac{P^2}{Q}d\mu$$

*and from Nishiyama and Sason [2020] we have*

$$KL(P\|Q) \leq \frac{1}{2}\left(TV(P\|Q) + \mathcal{X}^2(P\|Q)\right)\log e$$

*From Lemma .3 above we have*

$$GErr_t(f) \leq \frac{loge}{2}\mathbb{E}_{\mathbb{P}_{out}^{t,cov}}\left(TV(p(y_t|x_t)\|\mathcal{U})\right) + \frac{loge}{2}\mathbb{E}_{\mathbb{P}_{out}^{t,sem}}TV(F_f\|\mathcal{U})$$

$$+ \frac{loge}{2}d_{\mathcal{F}}(\mathbb{P}_{out}^{t.cov}, \mathbb{P}_{out}^{t,sem}) \quad (17)$$

$$+ \frac{loge}{2}\lambda + \frac{loge}{2}\mathbb{E}_{\mathbb{P}_{out}^{t,cov}}\left(\mathcal{X}^2(p(y_t|x_t)\|F_f(x_t))\right) + \mathbb{E}_{\mathbb{P}_{out}^{t,cov}}H(p(y_t|x_t)) \quad (18)$$

From Lemma .4 above we have

$$GErr_t(f) \leq \frac{loge}{2}\mathbb{E}_{\mathbb{P}_{out}^{t,cov}}\left(TV(p(y_t|x_t)\|\mathcal{U})\right) + \frac{loge}{2}\mathbb{E}_{\mathbb{P}_{out}^{t,sem}}\sqrt{\frac{1}{2}(\mathcal{L}_{reg}(f) - \log K)}$$

$$+ \frac{loge}{2}d_{\mathcal{F}}(\mathbb{P}_{out}^{t.cov}, \mathbb{P}_{out}^{t,sem}) + \frac{loge}{2}\lambda \quad (19)$$

$$+ \frac{loge}{2}\mathbb{E}_{\mathbb{P}_{out}^{t,cov}}\left(\mathcal{X}^2(p(y_t|x_t)\|F_f(x_t))\right) + \mathbb{E}_{\mathbb{P}_{out}^{t,cov}}H(p(y_t|x_t)) \quad (20)$$

since at each time $t$, $\mathbb{E}_{\mathbb{P}_{out}^{t,cov}}\left(TV(p(y_t|x_t)\|\mathcal{U})\right)$ is constant, we upper bound $GErr_t(f)$ as

$$GErr_t(f) \leq \frac{loge}{2}\mathbb{E}_{\mathbb{P}_{out}^{t,sem}}\sqrt{\frac{1}{2}(\mathcal{L}_{reg}(f) - \log K)} + \frac{loge}{2}d_{\mathcal{F}}(\mathbb{P}_{out}^{t.cov}, \mathbb{P}_{out}^{t,sem}) \quad (21)$$

$$+ C_t + \frac{loge}{2}\mathbb{E}_{\mathbb{P}_{out}^{t,cov}}\left(\mathcal{X}^2(p(y_t|x_t)\|F_f(x_t))\right) + \mathbb{E}_{\mathbb{P}_{out}^{t,cov}}H(p(y_t|x_t)) \quad (22)$$

**Lemma .6.** *Under the assumption* [**A2**] *and regularity condition on* $F_f^{\theta_1}$, *we have*

$$\mathbb{E}_{\mathbb{P}_{out}^{t,cov}}\left(\mathcal{X}^2(p(y_t|x_t)\|F_f(x_t))\right) \leq \delta_t^2 \, \mathbb{E}_{\mathbb{P}_{out}^{t,cov}}\left(I_F(\theta_2)\right) + B_t, \quad (23)$$

*where $I_F(\theta_2)$ is Fisher information and $B_t$ is constant. The key part of this conjecture is developed based on*

$$\mathbb{E}_{\mathbb{P}_{out}^{t,cov}}\left(\mathcal{X}^2(p(y_t|x_t)\|F_f(x_t))\right) = (\theta_1 - \theta_2)^2\mathbb{E}_{\mathbb{P}_{out}^{t,cov}}\left(I_F(\theta_2)\right) + o(\theta_1 - \theta_2)^2, \quad (24)$$

*where $\theta_1$ is approximately vanishes.*

Because inverse of entropy can be used as a confidence score to gauge the likelihood of a prediction being correct, we assume:
[**A3**] There exist a constant (say $Z_t$), such that

$$\mathbb{E}_{\mathbb{P}_{out}^{t+1,cov}}H(p(y_{t+1}|x_{t+1})) - \mathbb{E}_{\mathbb{P}_{out}^{t,cov}}H(p(y_t|x_t))) \geq Z_t + Conf_t - Conf_{t+1} \quad (25)$$

**Theorem 8.1. (Main Theorem)** *Let $\mathbb{P}^{t,cov}$ and $\mathbb{P}^{t,sem}$ be the covariate-shifted OOD and semantic OOD distribution. Denote $GErr_{t+1}(f)$ the generalization error at time $t$. Then at two time steps $t$ and $t+1$ and under assumptions* **[A1]** *and* **[A2]***, we have*

$$
GErr_{t+1}(f) - GErr_t(f) \geq -\tilde{\kappa}\, \Delta_{t\to t+1}^{cov,sem} - \tilde{\kappa}\, \Xi_{t\to t+1}^{sem} - \overline{\delta}_t^2\, \mathbb{E}_{\mathbb{P}_{out}^{t,cov}} (I_F(\theta_2))
$$
$$
+ C_{t\to t+1} + Conf_t - Conf_{t+1}, \tag{26}
$$

*where*

$$
\Delta_{t\to t+1}^{cov,sem} := d_{\mathcal{F}}(\mathbb{P}_{out}^{t+1,cov}, \mathbb{P}_{out}^{t+1,sem}) + d_{\mathcal{F}}(\mathbb{P}_{out}^{t,cov}, \mathbb{P}_{out}^{t,sem})
$$

*and*

$$
\Xi_{t\to t+1}^{sem} := \mathbb{E}_{\mathbb{P}_{out}^{t+1,sem}} \sqrt{\frac{1}{2}(\mathcal{L}_{reg}(f) - \log K)} + \mathbb{E}_{\mathbb{P}_{out}^{t,sem}} \sqrt{\frac{1}{2}(\mathcal{L}_{reg}(f) - \log K)}.
$$

*And $C_{t\to t+1} = C_{t+1} - C_t + B_t + Z_t$ and $\delta_t$ are constants and $\overline{\delta}_t^2 = \frac{loge}{2}\delta_t^2$.*

**Proof:** *Recall the definition of $GErr_t(f)$:*

$$
GErr_{t+1}(f) - GErr_t(f) \geq -\frac{1}{2\kappa}\mathbb{E}_{\mathbb{P}_{out}^{t+1,sem}} \sqrt{\frac{1}{2}(\mathcal{L}_{reg}(f) - \log K)} - \frac{1}{2\kappa}d_{\mathcal{F}}(\mathbb{P}_{out}^{t+1,cov}, \mathbb{P}_{out}^{t+1,sem})
$$
$$
- \frac{loge}{2}\mathbb{E}_{\mathbb{P}_{out}^{t,sem}} \sqrt{\frac{1}{2}(\mathcal{L}_{reg}(f) - \log K)} - \frac{loge}{2}d_{\mathcal{F}}(\mathbb{P}_{out}^{t.cov}, \mathbb{P}_{out}^{t,sem})
$$
$$
- \frac{loge}{2}\mathbb{E}_{\mathbb{P}_{out}^{t,cov}} \left( \mathcal{X}^2(p(y_t|x_t)\|F_f(x_t)) \right)
$$
$$
+ (C_{t+1} - C_t) + (\mathbb{E}_{\mathbb{P}_{out}^{t+1,cov}} H(p(y_{t+1}|x_{t+1})) - \mathbb{E}_{\mathbb{P}_{out}^{t,cov}} H(p(y_t|x_t))), \tag{27}
$$

*If we denote*

$$
\Delta_{t\to t+1}^{cov,sem} := d_{\mathcal{F}}(\mathbb{P}_{out}^{t+1,cov}, \mathbb{P}_{out}^{t+1,sem}) + d_{\mathcal{F}}(\mathbb{P}_{out}^{t,cov}, \mathbb{P}_{out}^{t,sem})
$$

*and*

$$
\Xi_{t\to t+1}^{sem} := \mathbb{E}_{\mathbb{P}_{out}^{t+1,sem}} \sqrt{\frac{1}{2}(\mathcal{L}_{reg}(f) - \log K)} + \mathbb{E}_{\mathbb{P}_{out}^{t,sem}} \sqrt{\frac{1}{2}(\mathcal{L}_{reg}(f) - \log K)},
$$

*then there exist a constant $\tilde{\kappa} \leq \frac{1}{2\kappa} + \frac{loge}{2}$ that (27) is written as*

$$
GErr_{t+1}(f) - GErr_t(f) \geq -\tilde{\kappa}\, \Delta_{t\to t+1}^{cov,sem} - \tilde{\kappa}\, \Xi_{t\to t+1}^{sem} + C_{t\to t+1}
$$
$$
- \frac{loge}{2}\mathbb{E}_{\mathbb{P}_{out}^{t,cov}} \left( \mathcal{X}^2(p(y_t|x_t)\|F_f(x_t)) \right)
$$
$$
+ \mathbb{E}_{\mathbb{P}_{out}^{t+1,cov}} H(p(y_{t+1}|x_{t+1})) - \mathbb{E}_{\mathbb{P}_{out}^{t,cov}} H(p(y_t|x_t))), \tag{28}
$$

*where $C_{t\to t+1} = C_{t+1} - C_t$ is constant. Apply the upper bound in Lemma .6, we have the lower bound below*

$$
GErr_{t+1}(f) - GErr_t(f) \geq -\tilde{\kappa}\, \Delta_{t\to t+1}^{cov,sem} - \tilde{\kappa}\, \Xi_{t\to t+1}^{sem} - \overline{\delta}_t^2\, \mathbb{E}_{\mathbb{P}_{out}^{t,cov}} (I_F(\theta_2))
$$
$$
+ C_{t\to t+1} + \mathbb{E}_{\mathbb{P}_{out}^{t+1,cov}} H(p(y_{t+1}|x_{t+1})) - \mathbb{E}_{\mathbb{P}_{out}^{t,cov}} H(p(y_t|x_t)), \tag{29}
$$

*where $C_{t\to t+1} = C_{t+1} - C_t + B_t$ is constant and $\overline{\delta}_t^2 = \frac{loge}{2}\delta_t^2$. By applying assumption* **[A3]***, we conclude the proof.*

## 9 Additional Experiments

**Evaluation Protocol.** Each model is evaluated after training on three separate test sets: the clean ID test set, the covariate-shifted test set, created by applying Gaussian noise to the ID data, and the semantic OOD test set. In our results, we report ID Acc. which is the accuracy on the clean ID test set, OOD Acc. which is the accuracy on the Gaussian-corrupted version of the test set, and finally FPR95, which is false positive rate when 95 percent of ID examples are correctly classified.

We compare TEMP-SCONE against the SCONE method, which serves as our primary baseline for OOD detection. SCONE is chosen for its strong performance in leveraging semantic consistency,

providing a relevant benchmark to evaluate the effectiveness of our approach. Note that all experiments are conducted using a consistent hardware setup with NVIDIA L40 GPUs. We ensure that both TEMP-SCONE and SCONE baselines are trained under the same conditions to provide a fair comparison.

A summary of the dynamic and distinct datasets used in our experiments is provided in Table 1 and Table 2.

| Experiment | ID Progression | Covariate Shift Applied | Semantic OOD Dataset(s) |
|---|---|---|---|
| Dynamic–CLEAR | CLEAR (10 sequential timesteps) (10 sequential timesteps) | Gaussian corrup (CLEAR-C) | LSUN-C, SVHN Places365 |
| Dynamic–YearBook | YearBook (7 temporal splits) | Gaussian corrup (YearBook-C) | FairFace |

Table 1: Experiment-oriented summary of dynamic datasets. Each experiment specifies the ID dataset progression, the covariate shift type applied, and the semantic OOD dataset(s) used.

| Experiment | ID Progression | Covariate Shift Applied | Semantic OOD Dataset(s) |
|---|---|---|---|
| Distinct–Exp 1 | CIFAR-10 → Imagenette → → CINIC-10 → STL-10 | Gaussian/Defocus corrup Gaussian/Defocus corrup | LSUN-C (all timesteps) LSUN-C (all timesteps) |
| Distinct–Exp 2 | CIFAR-10 → Imagenette → → CINIC-10 → STL-10 | Gaussian/Defocus corrup Gaussian/Defocus corrup | SVHN (all timesteps) SVHN (all timesteps) |
| Distinct–Exp 3 | CIFAR-10 → Imagenette → → CINIC-10 → STL-10 | Gaussian/Defocus corrup Gaussian/Defocus corrup | Places365 (all timesteps) Places365 (all timesteps) |
| Distinct–Exp 4 | CIFAR-10 → Imagenette → → CINIC-10 → STL-10 | Gaussian/Defocus corrup Gaussian/Defocus corrup | LSUN-C → SVHN → Places365 → DTD LSUN-C → SVHN → Places365 → DTD |

Table 2: Experiment-oriented summary of distinct datasets. Each experiment specifies the ID dataset progression, the covariate shift type applied, and the semantic OOD dataset(s) used. Note that Exp 4 is presented in main paper body.

We have executed additional experiments on both Gaussian noise and Defocus Blur covariate shifts on both dynamic and distinct dataset.

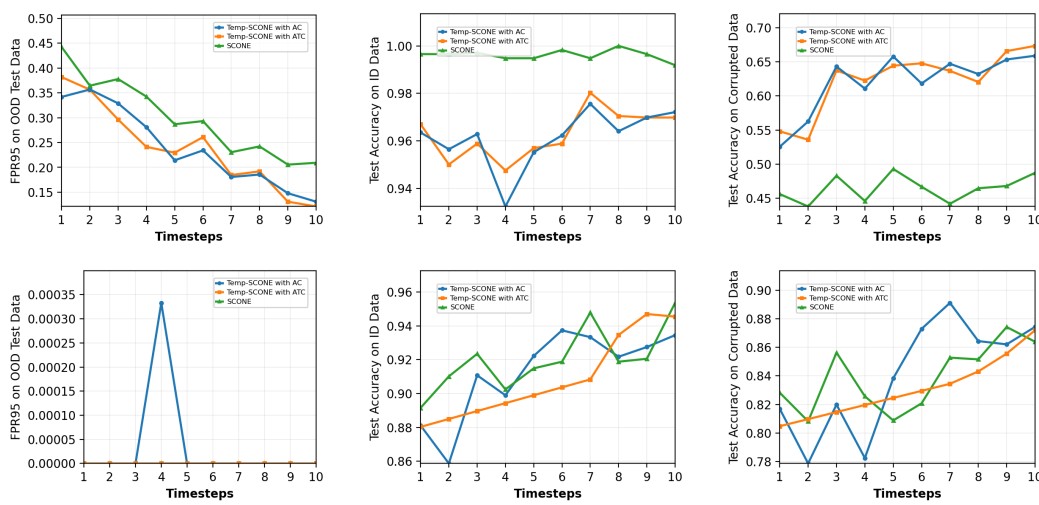

Figure 4: Dynamic Data (CLEAR - 10 timesteps), LSUN-C is OOD data, and Corruption type is Gaussian Noise (top row WRN, bottom row ViT). Columns show ID Acc.↑, OOD Acc.↑, FPR95 ↓.

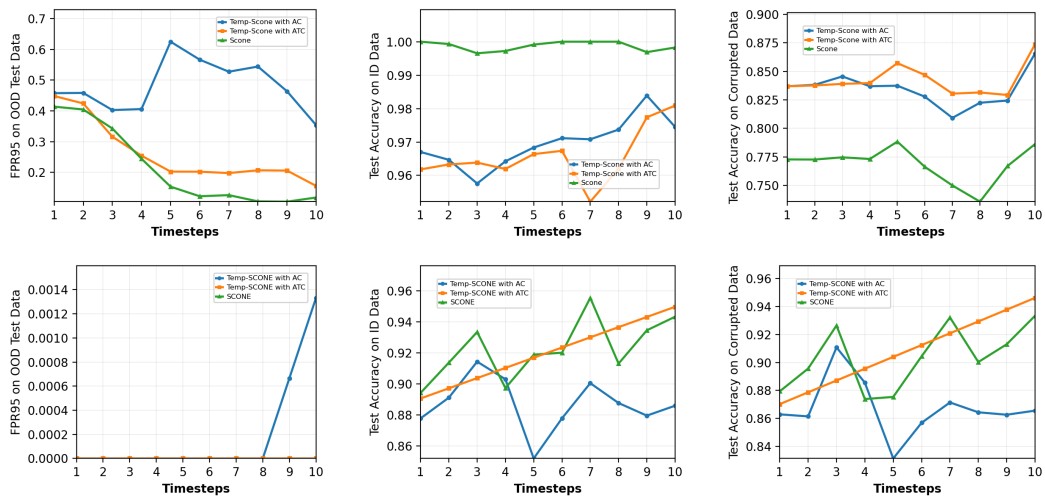

Figure 5: Dynamic Data (CLEAR - 10 timesteps), LSUN-C is OOD data, and Corruption type is Defocus Blur (top row WRN, bottom row ViT). Columns show ID Acc.↑, OOD Acc.↑, FPR95 ↓.

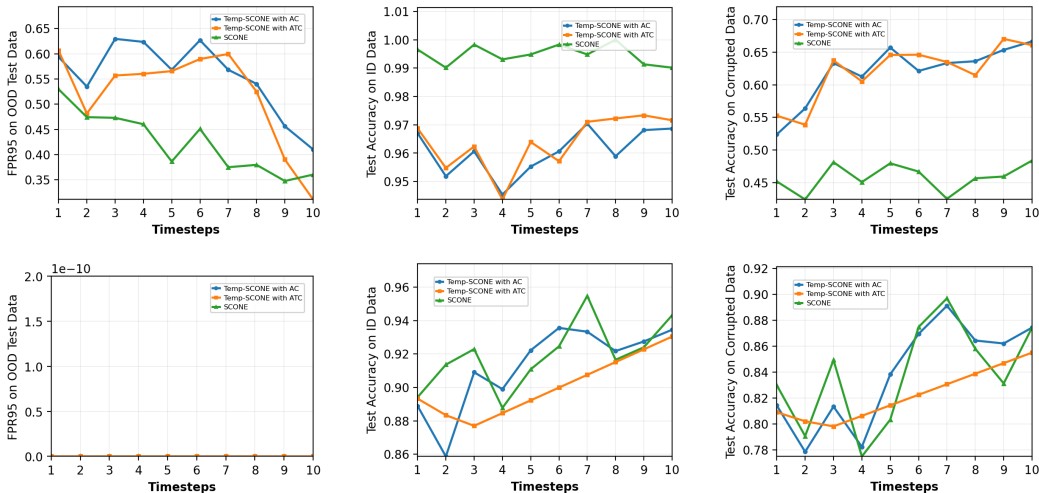

Figure 6: Dynamic Data (CLEAR - 10 timesteps), SVHN is OOD data, and Corruption type is Gaussian Noise (top row WRN, bottom row ViT). Columns show ID Acc.↑, OOD Acc.↑, FPR95 ↓.

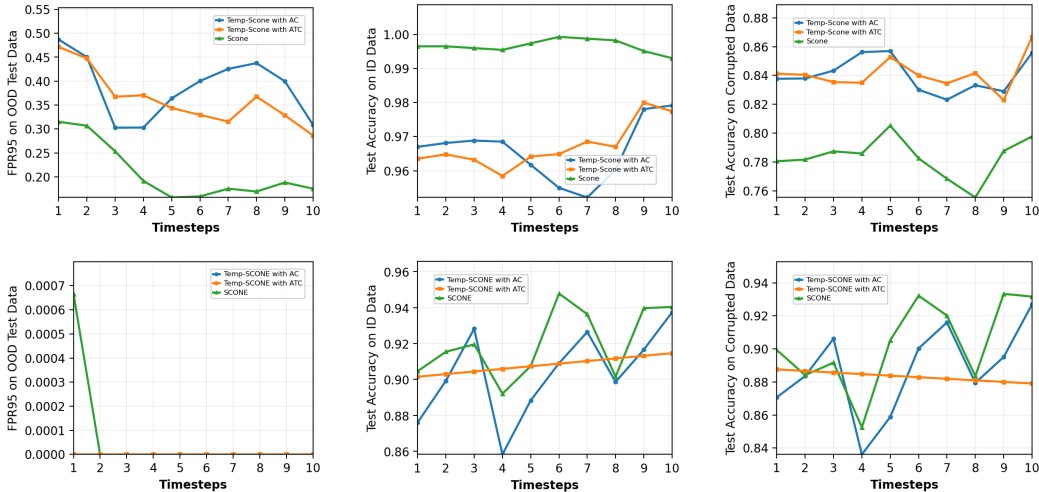

Figure 7: Dynamic Data (CLEAR - 10 timesteps), SVHN is OOD data, and Corruption type is Defocus Blur (top row WRN, bottom row ViT). Columns show ID Acc.↑, OOD Acc.↑, FPR95 ↓.

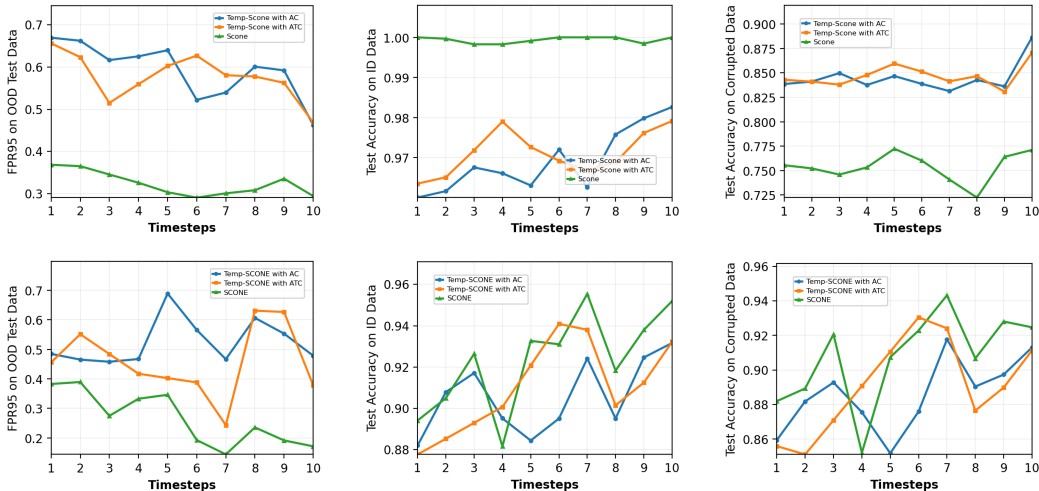

Figure 8: Dynamic Data (CLEAR - 10 timesteps), Places365 is OOD data, and Corruption type is Defocus Blur (top row WRN, bottom row ViT). Columns show ID Acc.↑, OOD Acc.↑, FPR95 ↓.

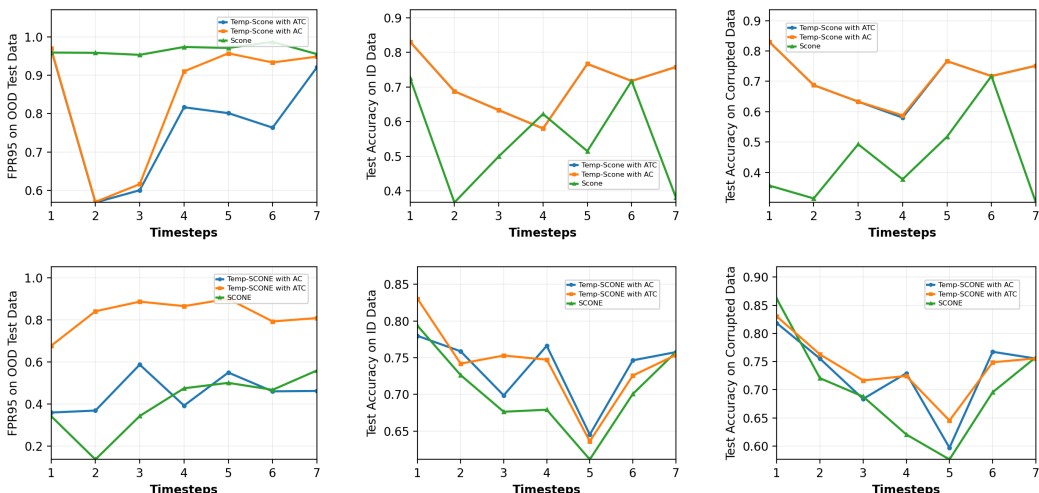

Figure 9: Dynamic Data (YearBook - 7 timesteps), FairFace is OOD data, and Corruption type is Defocus Blur (top row WRN, bottom row ViT). Columns show ID Acc.↑, OOD Acc.↑, FPR95 ↓.

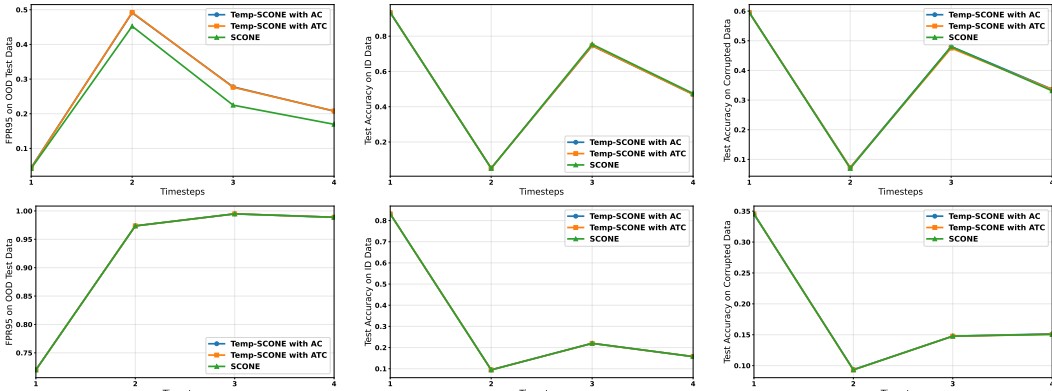

Figure 10: Distinct Data — Exp 1 (CIFAR-10 → Imagenette → CINIC-10 → STL-10 are four ID timesteps. Semantic OOD dataset is fixed as LSUN-C for all timesteps). Top row: WRN, bottom row: ViT. Columns show FPR95↓, ID test accuracy↑, and corrupted test accuracy↑.

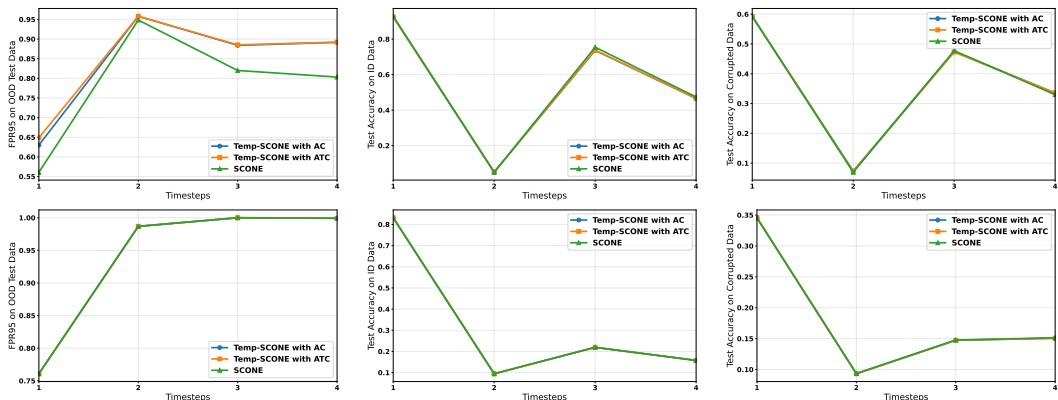

Figure 11: Distinct Data — Exp 2 (CIFAR-10 → Imagenette → CINIC-10 → STL-10 are the four ID timesteps; semantic OOD is fixed as SVHN). Top row: WRN, bottom row: ViT. Columns show FPR95↓, ID test accuracy↑, and corrupted test accuracy↑.

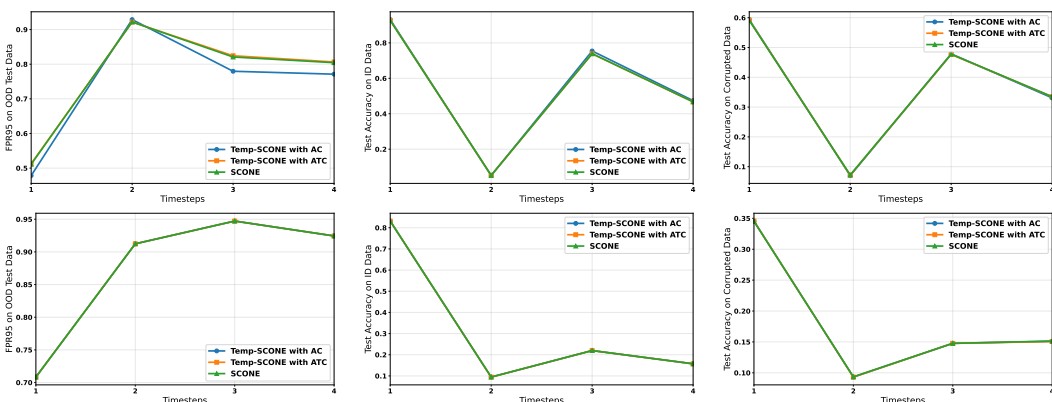

Figure 12: Distinct Data — Exp 3 (CIFAR-10 → Imagenette → CINIC-10 → STL-10 are the four ID timesteps; semantic OOD is fixed as Places365). Top row: WRN, bottom row: ViT. Columns show FPR95↓, ID test accuracy↑, and corrupted test accuracy↑.

Table 3: Distinct Datasets Full experimental results across experiments, models, and methods with Gaussian Noise corruption. Each row reports FPR95, ID accuracy, and corrupted accuracy at Steps 1–2.

| Exp | Model | Method | Step 1 (FPR) | Step 1 (ID Acc) | Step 1 (OOD Acc) | Step 2 (FPR) | Step 2 (ID Acc) | Step 2 (OOD Acc) |
|---|---|---|---|---|---|---|---|---|
| 1 | WRN | SCONE | 4.24 | 93.49 | 59.57 | 45.24 | 5.02 | 6.97 |
| 1 | WRN | Temp-SCONE ATC | 4.36 | 93.41 | 59.53 | 49.24 | 5.18 | 7.21 |
| 1 | WRN | Temp-SCONE AC | 4.48 | 93.29 | 59.49 | 49.12 | 5.18 | 7.25 |
| 1 | ViT | SCONE | 72.00 | 83.16 | 34.56 | 97.36 | 9.46 | 9.31 |
| 1 | ViT | Temp-SCONE ATC | 72.00 | 83.16 | 34.56 | 97.36 | 9.46 | 9.31 |
| 1 | ViT | Temp-SCONE AC | 72.00 | 83.16 | 34.52 | 97.36 | 9.46 | 9.35 |
| 2 | WRN | SCONE | 56.08 | 92.87 | 59.25 | 94.84 | 4.98 | 6.93 |
| 2 | WRN | Temp-SCONE ATC | 64.88 | 92.44 | 59.30 | 95.80 | 5.10 | 7.25 |
| 2 | WRN | Temp-SCONE AC | 63.00 | 92.64 | 59.26 | 95.80 | 5.10 | 7.33 |
| 2 | ViT | SCONE | 76.08 | 83.16 | 34.52 | 98.68 | 9.46 | 9.35 |
| 2 | ViT | Temp-SCONE ATC | 76.08 | 83.16 | 34.56 | 98.68 | 9.46 | 9.31 |
| 2 | ViT | Temp-SCONE AC | 76.08 | 83.16 | 34.56 | 98.68 | 9.46 | 9.31 |
| 3 | WRN | SCONE | 51.12 | 92.67 | 59.10 | 92.20 | 5.26 | 7.21 |
| 3 | WRN | Temp-SCONE ATC | 51.28 | 92.71 | 59.20 | 92.16 | 5.22 | 7.17 |
| 3 | WRN | Temp-SCONE AC | 47.84 | 93.14 | 59.37 | 92.88 | 5.10 | 7.01 |
| 3 | ViT | SCONE | 70.84 | 83.16 | 34.52 | 91.24 | 9.46 | 9.35 |
| 3 | ViT | Temp-SCONE ATC | 70.84 | 83.16 | 34.56 | 91.24 | 9.46 | 9.31 |
| 3 | ViT | Temp-SCONE AC | 70.84 | 83.16 | 34.56 | 91.24 | 9.46 | 9.31 |
| 4 | WRN | SCONE | 4.52 | 93.18 | 59.25 | 95.96 | 5.10 | 7.29 |
| 4 | WRN | Temp-SCONE ATC | 4.24 | 93.41 | 59.41 | 96.00 | 5.06 | 7.33 |
| 4 | WRN | Temp-SCONE AC | 4.24 | 93.41 | 59.80 | 95.72 | 5.06 | 7.08 |
| 4 | ViT | SCONE | 72.00 | 83.16 | 34.52 | 98.68 | 9.46 | 9.35 |
| 4 | ViT | Temp-SCONE ATC | 72.00 | 83.16 | 34.56 | 98.68 | 9.46 | 9.31 |
| 4 | ViT | Temp-SCONE AC | 72.00 | 83.16 | 34.56 | 98.68 | 9.46 | 9.31 |

Table 4: Distinct Datasets Full experimental results across experiments, models, and methods with Gaussian Noise corruption. Each row reports FPR95, ID accuracy, and corrupted accuracy at Steps 3–4.

| Exp | Model | Method | Step 3 (FPR) | Step 3 (ID Acc) | Step 3 (OOD Acc) | Step 4 (FPR) | Step 4 (ID Acc) | Step 4 (OOD Acc) |
|---|---|---|---|---|---|---|---|---|
| 1 | WRN | SCONE | 22.48 | 75.42 | 48.01 | 16.96 | 47.52 | 33.13 |
| 1 | WRN | Temp-SCONE ATC | 27.64 | 74.61 | 47.51 | 20.80 | 46.97 | 33.57 |
| 1 | WRN | Temp-SCONE AC | 27.76 | 74.57 | 48.05 | 20.76 | 47.05 | 33.64 |
| 1 | ViT | SCONE | 99.48 | 22.01 | 14.77 | 98.88 | 15.73 | 15.06 |
| 1 | ViT | Temp-SCONE ATC | 99.48 | 21.98 | 14.77 | 98.88 | 15.73 | 15.06 |
| 1 | ViT | Temp-SCONE AC | 99.48 | 21.98 | 14.77 | 98.88 | 15.73 | 15.14 |
| 2 | WRN | SCONE | 82.00 | 75.42 | 47.73 | 80.32 | 47.40 | 32.98 |
| 2 | WRN | Temp-SCONE ATC | 88.52 | 73.78 | 47.24 | 89.20 | 46.58 | 33.63 |
| 2 | WRN | Temp-SCONE AC | 88.40 | 73.63 | 47.51 | 89.16 | 46.58 | 33.45 |
| 2 | ViT | SCONE | 100.00 | 21.94 | 14.77 | 99.92 | 15.73 | 15.14 |
| 2 | ViT | Temp-SCONE ATC | 100.00 | 21.94 | 14.73 | 99.92 | 15.73 | 15.06 |
| 2 | ViT | Temp-SCONE AC | 100.00 | 21.94 | 14.73 | 99.92 | 15.81 | 15.06 |
| 3 | WRN | SCONE | 82.08 | 73.89 | 47.75 | 80.44 | 46.74 | 33.48 |
| 3 | WRN | Temp-SCONE ATC | 82.44 | 73.85 | 47.67 | 80.64 | 46.70 | 33.49 |
| 3 | WRN | Temp-SCONE AC | 77.96 | 75.42 | 47.85 | 77.12 | 47.40 | 33.10 |
| 3 | ViT | SCONE | 94.72 | 21.98 | 14.77 | 92.44 | 15.73 | 15.14 |
| 3 | ViT | Temp-SCONE ATC | 94.72 | 21.98 | 14.77 | 92.44 | 15.69 | 15.06 |
| 3 | ViT | Temp-SCONE AC | 94.72 | 21.98 | 14.77 | 92.44 | 15.73 | 15.10 |
| 4 | WRN | SCONE | 82.76 | 73.66 | 47.90 | 88.84 | 46.54 | 33.37 |
| 4 | WRN | Temp-SCONE ATC | 83.04 | 73.81 | 48.05 | 89.01 | 46.47 | 33.57 |
| 4 | WRN | Temp-SCONE AC | 80.12 | 74.84 | 48.28 | 87.42 | 47.05 | 33.07 |
| 4 | ViT | SCONE | 94.72 | 21.98 | 14.77 | 98.64 | 15.69 | 15.14 |
| 4 | ViT | Temp-SCONE ATC | 94.72 | 21.98 | 14.77 | 98.64 | 15.69 | 15.06 |
| 4 | ViT | Temp-SCONE AC | 94.72 | 21.98 | 14.77 | 98.64 | 15.69 | 15.06 |

Table 5: Distinct Datasets Full experimental results across experiments, models, and methods with Defocus Blur. Each row reports FPR95, ID accuracy, and corrupted accuracy at Steps 1–2.

| Exp | Model | Method | Step 1 (FPR) | Step 1 (ID Acc) | Step 1 (OOD Acc) | Step 2 (FPR) | Step 2 (ID Acc) | Step 2 (OOD Acc) |
|---|---|---|---|---|---|---|---|---|
| 1 | WRN | SCONE | 3.72 | 94.23 | 73.72 | 43.24 | 5.06 | 8.76 |
| 1 | WRN | Temp-SCONE ATC | 4.48 | 94.23 | 72.60 | 45.48 | 5.28 | 8.80 |
| 1 | WRN | Temp-SCONE AC | 4.40 | 94.12 | 72.80 | 45.60 | 5.32 | 8.80 |
| 1 | ViT | SCONE | 72.00 | 83.16 | 81.91 | 97.36 | 9.46 | 9.19 |
| 1 | ViT | Temp-SCONE ATC | 72.00 | 83.16 | 81.91 | 97.36 | 9.46 | 9.19 |
| 1 | ViT | Temp-SCONE AC | 72.00 | 83.16 | 81.91 | 97.36 | 9.46 | 9.19 |
| 2 | WRN | SCONE | 65.04 | 93.26 | 74.42 | 96.08 | 5.16 | 8.68 |
| 2 | WRN | Temp-SCONE ATC | 65.28 | 93.18 | 74.47 | 96.00 | 5.16 | 8.72 |
| 2 | WRN | Temp-SCONE AC | 50.00 | 93.33 | 74.69 | 92.80 | 5.17 | 8.64 |
| 2 | ViT | SCONE | 76.08 | 83.16 | 81.91 | 98.68 | 9.46 | 9.19 |
| 2 | ViT | Temp-SCONE ATC | 76.08 | 83.16 | 81.91 | 98.68 | 9.46 | 9.19 |
| 2 | ViT | Temp-SCONE AC | 76.08 | 83.16 | 81.91 | 98.68 | 9.46 | 9.19 |
| 3 | WRN | SCONE | 51.08 | 93.57 | 71.57 | 92.16 | 5.06 | 9.27 |
| 3 | WRN | Temp-SCONE ATC | 52.92 | 93.53 | 71.04 | 92.60 | 5.17 | 8.92 |
| 3 | WRN | Temp-SCONE AC | 51.72 | 93.57 | 70.92 | 92.56 | 5.21 | 8.96 |
| 3 | ViT | SCONE | 70.84 | 83.16 | 81.91 | 91.24 | 9.46 | 9.19 |
| 3 | ViT | Temp-SCONE ATC | 70.84 | 83.16 | 81.91 | 91.24 | 9.46 | 9.19 |
| 3 | ViT | Temp-SCONE AC | 70.84 | 83.16 | 81.91 | 91.24 | 9.46 | 9.19 |
| 4 | WRN | SCONE | 4.28 | 94.15 | 72.99 | 96.36 | 5.12 | 8.65 |
| 4 | WRN | Temp-SCONE ATC | 4.44 | 94.15 | 72.84 | 96.40 | 5.05 | 8.68 |
| 4 | WRN | Temp-SCONE AC | 3.80 | 94.19 | 73.45 | 94.56 | 5.13 | 8.53 |
| 4 | ViT | SCONE | 72.00 | 83.16 | 81.91 | 98.68 | 9.46 | 9.19 |
| 4 | ViT | Temp-SCONE ATC | 72.00 | 83.16 | 81.91 | 98.68 | 9.46 | 9.19 |
| 4 | ViT | Temp-SCONE AC | 72.00 | 83.16 | 81.91 | 98.68 | 9.46 | 9.19 |

Table 6: Distinct Datasets Full experimental results across experiments, models, and methods with Defocus Blur. Each row reports FPR95, ID accuracy, and corrupted accuracy at Steps 3–4.

| Exp | Model | Method | Step 3 (FPR) | Step 3 (ID Acc) | Step 3 (OOD Acc) | Step 4 (FPR) | Step 4 (ID Acc) | Step 4 (OOD Acc) |
|---|---|---|---|---|---|---|---|---|
| 1 | WRN | SCONE | 18.92 | 76.50 | 56.22 | 16.08 | 47.91 | 38.79 |
| 1 | WRN | Temp-SCONE ATC | 23.12 | 75.62 | 55.95 | 19.08 | 47.29 | 38.44 |
| 1 | WRN | Temp-SCONE AC | 23.20 | 75.58 | 55.71 | 18.88 | 47.37 | 38.28 |
| 1 | ViT | SCONE | 99.48 | 21.90 | 20.62 | 98.88 | 15.73 | 14.53 |
| 1 | ViT | Temp-SCONE ATC | 99.48 | 21.94 | 20.62 | 98.88 | 15.73 | 14.53 |
| 1 | ViT | Temp-SCONE AC | 99.48 | 21.90 | 20.62 | 98.88 | 15.73 | 14.57 |
| 2 | WRN | SCONE | 87.80 | 74.53 | 58.75 | 85.92 | 46.89 | 39.53 |
| 2 | WRN | Temp-SCONE ATC | 87.80 | 74.49 | 58.75 | 86.04 | 46.89 | 39.53 |
| 2 | WRN | Temp-SCONE AC | 80.72 | 75.80 | 58.71 | 75.12 | 47.56 | 39.65 |
| 2 | ViT | SCONE | 100.00 | 21.90 | 20.66 | 99.92 | 15.69 | 14.53 |
| 2 | ViT | Temp-SCONE ATC | 100.00 | 21.90 | 20.66 | 99.92 | 15.69 | 14.53 |
| 2 | ViT | Temp-SCONE AC | 100.00 | 21.90 | 20.66 | 99.92 | 15.69 | 14.57 |
| 3 | WRN | SCONE | 79.20 | 76.12 | 55.09 | 78.68 | 47.95 | 37.89 |
| 3 | WRN | Temp-SCONE ATC | 81.68 | 75.19 | 54.97 | 81.08 | 47.13 | 37.59 |
| 3 | WRN | Temp-SCONE AC | 81.44 | 75.00 | 54.78 | 81.36 | 47.25 | 37.43 |
| 3 | ViT | SCONE | 94.76 | 21.90 | 20.58 | 92.52 | 15.77 | 14.53 |
| 3 | ViT | Temp-SCONE ATC | 94.76 | 21.90 | 20.58 | 92.52 | 15.77 | 14.53 |
| 3 | ViT | Temp-SCONE AC | 94.76 | 21.86 | 20.58 | 92.52 | 15.77 | 14.53 |
| 4 | WRN | SCONE | 82.40 | 74.99 | 55.33 | 88.13 | 47.25 | 37.66 |
| 4 | WRN | Temp-SCONE ATC | 82.16 | 74.99 | 55.17 | 88.25 | 47.13 | 37.55 |
| 4 | WRN | Temp-SCONE AC | 79.28 | 75.92 | 55.56 | 86.47 | 47.76 | 37.97 |
| 4 | ViT | SCONE | 94.76 | 21.86 | 20.58 | 98.70 | 15.77 | 14.53 |
| 4 | ViT | Temp-SCONE ATC | 94.72 | 21.90 | 20.58 | 98.70 | 15.77 | 14.53 |
| 4 | ViT | Temp-SCONE AC | 94.72 | 21.86 | 20.58 | 98.70 | 15.77 | 14.53 |

