# OpenReview forum: "Temp-SCONE: A Novel Out-of-Distribution Detection and Domain Generalization Framework for Wild Data with Temporal Shift"
_NeurIPS.cc/2025/Workshop/Reliable_ML — NeurIPS 2025 - Reliable ML Workshop_

### Official Review · Reviewer_wKfg · 2025-09-11
**Extending SCONE for Temporal Shifts: Useful but Incremental**

**Rating:** 5
**Confidence:** 4

**Review:**

## Brief Summary:
This paper extends the SCONE method, which is designed to address domain generalization (DG) under covariate shift and to differentiate DG caused by semantic shift. However, SCONE is limited to static environments because it does not account for temporal shifts. The proposed method addresses this limitation by incorporating an additional regularization term that penalizes decreasing prediction confidence, thereby explicitly modeling temporal dynamics.

## Strengths. Novelty, rigor, empirical/theoretical quality, clarity, relevance to reliability with imperfect data.
The motivation is valid, the method is well demonstrated and it suits the theme of the workshop that handle the imperfect data that drift with time.
## Weaknesses / Limitations. Missing comparisons/ablations, unclear assumptions, proof gaps, failure modes, scope limits.
1. From my perspective, this work appears to be incremental.
2. In the experiments, the authors simulate covariate shift by adding Gaussian noise. However, this approach seems overly simplistic, as covariate shift is not equivalent to merely injecting Gaussian noise.
3. Some experimental results are not clearly explained. For example, in Figure 2 (top row, right subfigure), why does SCONE outperform the proposed method on a dynamic dataset? Furthermore, in many dynamic datasets, SCONE’s performance does not exhibit the expected monotonic decline which also needs clarification.
4. The writing is occasionally redundant. For instance, lines 168–172 repeat descriptions of the dataset settings, which could be streamlined for clarity.
## Suggestions for Authors. Specific things that would improve the paper:
The authors employ the Augmented Lagrangian Method (ALM) to impose the temporal constraint. It would be beneficial for them to illustrate the dynamics of each term in their loss function to help readers better understand the training process. For example, visualizing how each term evolves during optimization would clarify which term dominates at different stages, especially considering that the Lagrange multiplier can become very large as optimization progresses.
## Ethics (if applicable). Note any concerns (about privacy, fairness, misuse, sensitive data use) and suggested mitigations.
N/A

---

### Official Review · Reviewer_fNAv · 2025-09-19
**Temp-SCONE: A Novel Out-of-Distribution Detection and Domain Generalization Framework for Wild Data with Temporal Shift**

**Rating:** 8
**Confidence:** 4

**Review:**

The setting of OWL (open-world learning) is a commonly encountered problem setting where models are pre-trained and then adjusted in an unsupervised fashion; in the specific case of using SCONE, such adjustment occurs with the use of an energy loss term that "pushes away" those data points that appear to be semantic shifts based on the trained classifier. Over this OWL adapation, however, the classifier can commonly change its behavior quite dramatically, which will often arise due to temporal shifts in the $P_{wild}$ distribution. For this reason, we want to additionally enforce consistency of the classifier performance, as measured by the in-distribution average confidence (on the true class), over time.

The idea is quite intuitive and seems like a worthwhile addition to the general SCONE framework. I had a bit of a tough time parsing the generalization error statement, although the paragraph after did a decent job explaining the intuition. Overall, this seems like a strong contribution to the robustness space.